# Incidence and Risk Factors of Reinfection with HCV after Treatment in People Living with HIV

**DOI:** 10.3390/v14020439

**Published:** 2022-02-21

**Authors:** Chien-Yu Cheng, Shin-Yen Ku, Yi-Chun Lin, Cheng-Pin Chen, Shu-Hsing Cheng, I-Feng Lin

**Affiliations:** 1Department of Infectious Diseases, Taoyuan General Hospital, Ministry of Health and Welfare, Taoyuan City 330, Taiwan; s841060@yahoo.com (C.-Y.C.); jean640514@gmail.com (Y.-C.L.); jangbin@gmail.com (C.-P.C.); shuhsingcheng@gmail.com (S.-H.C.); 2Institute of Public Health, School of Medicine, National Yang-Ming Chiao Tung University, Taipei City 112, Taiwan; 3Department of Nursing, Taoyuan General Hospital, Ministry of Health and Welfare, Taoyuan City 330, Taiwan; yen376@gmail.com; 4School of Clinical Medicine, National Yang-Ming Chiao Tung University, Taipei City 112, Taiwan; 5School of Public Health, Taipei Medical University, Taipei City 110, Taiwan

**Keywords:** people who inject drugs (PWID), reinfection with HCV, people living with HIV (PLWH), heroin dependency, amphetamine abuse, men who have sex with men (MSM)

## Abstract

Infection with hepatitis C virus (HCV) does not induce protective immunity, and re-exposure to HCV can reinfect the population engaging in high-risk behavior. An increasing incidence of acute hepatitis C infection in people living with HIV (PLWH) has been described in recent years. This retrospective cohort study was conducted in PLWH who completed HCV therapy between June 2009 and June 2020 at an HIV care hospital, to analyze their basic characteristics and risky behavior. Of 2419 patients, 639 were diagnosed with HCV infection and 516 completed the HCV therapy with a sustained virologic response. In total, 59 patients (11.4%) were reinfected with acute hepatitis C, and the median time to reinfection was 85.3 weeks (IQR: 57–150). The incidence of reinfection was 6.7 cases/100 person-years. The factors associated with reinfection were being male (AHR, 8.02; 95% CI 1.08–59.49), DAA (direct-acting antiviral) treatment (AHR, 2.23; 95% CI 1.04–4.79), liver cirrhosis (AHR, 3.94; 95% CI 1.09–14.22), heroin dependency (AHR: 7.41; 95% CI 3.37–14.3), and HIV viral loads <50 copies/mL at the follow-up (AHR: 0.47, 95% CI 0.24–0.93) in the subgroup of people who inject drugs (PWID). Amphetamine abuse (AHR: 20.17; 95% CI 2.36–172.52) was the dominant factor in the subgroup of men who have sex with men (MSM). Our study suggests that education and behavioral interventions are needed in this population to prevent reinfection.

## 1. Introduction

An estimated 71 million people have chronic HCV infection, and a significant number of these people develop cirrhosis of the liver or liver cancer as a result [1]. Annually, 1.34 million deaths are caused by viral hepatitis, which is comparable to the deaths caused by tuberculosis and higher than those by HIV infection. HCV infection remains a major cause of liver-related morbidity and mortality among people living with HIV (PLWH), especially the people who inject drugs (PWID) [2,3]. The treatment uptake with interferon-based therapy has generally been high in the population of HIV-positive men who have sex with men (MSM), and high sustained virological response (SVR) rates have been reported, but there are relatively low SVR rates in the population of PWID [4]. Fortunately, oral direct antiviral agent (DAA)-based treatment trials have demonstrated very high efficacy (SVR > 95%) among PLWH with HCV coinfection, even in the population of PWID [5,6]. Hence, some studies have predicted that scaling-up DAA treatment will substantially reduce the HCV prevalence [7,8].

An increasing incidence of HCV reinfection has been documented among MSM living with HIV in studies since 2010 [9,10,11,12]. A meta-analysis also reported that the overall HCV reinfection rate is high in PWID who continued their illicit drug use without opioid substitution therapy (OST) [13]. The most common risk factors for HCV infection included the sharing of needles for heroin injection and engaging in condomless anal intercourse with multiple sexual partners, though the interaction between behavioral factors and immunological factors associated with HCV reinfection is little known. Most existing research has put forward findings based on small sample sizes, a short duration of observation, and single-arm studies. In Europe, the duration of follow-up of studies after SVR ranges from four to eight years in general, and no such study has been conducted in Asia [14,15,16]. So, in this research, we assessed the incidence of HCV reinfection after a sustained virological response and examined the behavioral and immunological risk factors that played a role in affecting the outcomes in the observed cohort.

## 2. Materials and Methods

We conducted this observational cohort study at an HIV care hospital from June 2009 to June 2020. Briefly, people living with HIV who were diagnosed with acute or chronic HCV infection were included in the cohort, and clinical and virological data on their basic characteristics were retrospectively collected for both the primary HIV and their HCV infection. In Taiwan, face-to-face interviews assessing the patients’ illicit drug abuse (such as heroin, amphetamines, and methadone) and risky sexual behavior were performed by HIV case managers when the patients returned for new rounds of their highly active anti-retroviral therapy every three months at outpatient clinics. If acute HCV infection was diagnosed during the follow-up, clinical covariates were taken as the closest measurements to the date of follow-up, including the CD4 lymphocyte count, HIV RNA viral load, HCV RNA viral load, HCV genotype, regimens of HCV treatment, and STDs, with risky behavior assumed in the past three months.

### 2.1. Study Procedure

Systemic screening for the HCV antibody was performed annually in all HIV-positive patients between 1 June 2009 and 31 December 2019. The HCV RNA was tested for in-patients who were positive for the HCV antibody, and treatment consisted of peginterferon alfa plus ribavirin (PegIFN/RBV), or DAAs with or without ribavirin. The HCV RNA was tested at the baseline before HCV therapy, at the end of treatment, at follow-up week 12 or 24, and then at least once annually. A sustained virological response (SVR) was defined as one negative HCV RNA test 12 or 24 weeks after the end of treatment (depending on the treatment regimen). Patients were tested for the presence of HCV RNA every year at their physician’s discretion, with additional testing if HCV reinfection was suspected.

The following covariates were included in the model: age, gender, risk factors for HCV and HIV infection, CD4 lymphocyte count, HIV RNA viral load before and after HCV therapy, HCV RNA viral load, HCV genotype, regimens of HCV therapy, risky behavior, STDs, HCV treatment year, and history of liver cirrhosis.

Demographic information on each patient’s age, gender (male or female), and risk of HCV infection were obtained from their medical records, and their CD4 lymphocyte count, HIV RNA viral load, HCV RNA viral load, and HCV genotype were acquired from medical reports at the hospital. The HCV genotype was classified as either 1, 2, 3, 4, or 6, and a CD4 lymphocyte count above 500 cells/mL was defined as adequate immunity. The HIV RNA was measured with COBAS AmpliPrep TaqMan HIV-1 test version 2.0 (Roche, Mannheim, Germany), with a lower limit of quantitation of fewer than 20 copies/mL. HIV RNA below 50 copies/mL was defined as an HIV status that was under control.

Information about risky behavior was obtained from HIV case managers’ extensive counseling of patients during their follow-up every three months. Risky behavior included heroin dependency, methadone replacement therapy, amphetamine abuse, and multiple sexual partners.

### 2.2. Definition

HCV reinfection was defined as a positive HCV RNA test, preceded by at least one negative HCV RNA test result after HCV treatment. The estimated date of reinfection was the date of the first positive test result after the end of treatment. Plasma HCV RNA viral loads were measured using a Roche LightCycler RNA Master SYBR Green I/High Pure Viral Nucleic Acid Kit (Roche, Mannheim, Germany) with a lower limit of quantitation of less than 10 IU/mL. The HCV genotype was determined using an Abbott HCV Real-Time Genotype II assay (Abbott Diagnostics, Chicago, IL, USA).

### 2.3. Incidence

Follow-up began at the end of treatment and continued until the estimated date of HCV reinfection for those who had reinfection, or the last HIV follow-up and negative HCV RNA test for those who did not experience reinfection. Patients who completed their HCV treatment then became at risk of subsequent reinfection. The incidence rate (IR) of reinfection and its corresponding 95% confidence interval (CI) were calculated using a person–time approach.

### 2.4. Statistical Analysis

Patients’ baseline characteristics are presented using descriptive statistics: the mean ± standard deviation or median ± interquartile range for continuous variables, and the frequency (percentage) for categorical variables. Categorical data were analyzed using χ^2^ or Fisher’s exact test, as appropriate, and continuous variables were compared using the Mann–Whitney U test. The 95% CI of the hazard ratio was computed using a binomial distribution.

### 2.5. Analysis and Cox Proportional Hazard Models

The cumulative incidence of the first reinfection during follow-up was estimated using the Kaplan–Meier test. Univariate and multivariate Cox proportional hazard models were generated to evaluate the risk factors of reinfection. The assumption of proportional hazards was tested using Schoenfeld residuals test. Certain important variables (*p* < 0.1 in χ^2^, Fisher’s exact test, or univariate analysis) were selected for subsequent multivariate analysis. All analyses were two-tailed, and *p* < 0.05 was considered statistically significant. All analyses were conducted using SPSS version 24.0.

## 3. Results

### 3.1. Baseline Characteristics

We identified 516 HIV-positive patients with a documented HCV infection cure between June 2009 and June 2020, as shown in Figure 1. The baseline characteristics of the participants are presented in Table 1. The majority of the included patients were male (78.9%) with a mean age of 42.0 years. The study population consisted mainly of PWID (78.9%), followed by men who have sex with men (MSM, 20.3%), and heterosexuals (0.8%). Most of the patients (98.4%) had received ART at the time of HCV therapy initiation, with a median CD4 count of 506 cells/mm^3^ (IQR 369–692), and 89.7% had plasma HIV RNA < 50 copies/mL. Before HCV therapy initiation, the median plasma HCV RNA load was 6.32 log_10_ IU/mL, and the dominant genotype was genotype 6 (n = 176, 34.1%) followed by 1b (n = 117, 22.7%), 1a (n = 101, 9.6%), 2 (n = 82, 15.9%), and 3 (n = 44, 8.5%). A SVR following treatment with pegylated interferon with/without ribavirin was achieved in 129 (25%) patients, while 387 (75%) patients were treated with DAAs (Table 1).

### 3.2. Immunological Status and Counseling Interview at Follow-Up

Through extensive counseling, 167 patients (32.4%) were found to have taken methadone replacement therapy, 142 (27.5%) were heroin dependent, 71 (13.8%) abused amphetamine, 38 (7.4%) had multiple sexual partners, and 35 (6.8%) were diagnosed with syphilis in the past three months. Finally, 59 of these patients (11.4%) presented with an episode of acute HCV reinfection, and the median duration of HCV reinfection was 85.3 weeks (IQR 4.8–12.5). The median follow-up time for the 516 patients we included was 63.6 weeks (IQR 48.3–114.5), totaling 884.6 person-years. A total of 59 reinfections occurred, and the overall incidence rate of HCV reinfection was 6.7/100 person-years (Table 2).

### 3.3. Risk Factors of HCV Reinfection: Overall and Subgroup Analyses

As shown in Table 3 HCV reinfection was analyzed in Cox proportional hazards regression model, and the factors associated with reinfection were liver cirrhosis (AHR 3.61, 95% CI 1.02–12.71), heroin dependency (AHR 9.35, 95% CI 4.28–20.44), and multiple sexual partners (AHR 56.37, 95% CI 6.12–519.53). In the subgroup of PWID (Table 4), being male (AHR, 8.02; 95% CI 1.08–59.49), DAA (direct-acting antiviral) treatment (AHR, 2.23; 95% CI 1.04–4.79), liver cirrhosis (AHR, 3.94; 95% CI 1.09–14.22), and heroin dependency (AHR: 7.41, 95% CI 3.37–14.3) were risk factors, while an HIV viral load <50 copies/mL at the follow-up (AHR: 0.47, 95% CI 0.24–0.93) was a protective factor against reinfection. Then, in the subgroup of MSM (Table 5), amphetamine abuse (AHR: 20.17, 95% CI 2.36–172.52) was the only risk factor.

Figure 2 shows the trends of HCV reinfection in the subgroups of PWID with or without heroin dependency and MSM with or without amphetamine abuse using Kaplan–Meier curves.

## 4. Discussion

In this study of 516 PLWH with chronic or acute HCV infection, we assessed the incidence and risk factors of HCV reinfection among those who completed DAA or peginterferon treatment with an SVR. During a median follow-up of 1.2 years, we found an incidence of 6.7/100 person-years. In Cox proportional hazards regression model, the factors associated with HCV reinfection were being male, DAA treatment, liver cirrhosis, and heroin dependency in the subgroup of PWID, and amphetamine abuse was the dominant factor in the subgroup of MSM. To date, our observational cohort study spanning more than 10 years is the first, to our knowledge, that assesses behavioral risk factors for HCV reinfection in Asia.

In PWID, the incidence of HCV reinfection ranged between 0.8 and 33.0 per 100PY in the findings of different small studies or cohorts [17,18,19,20], and the pooled reinfection incidence in injected drug users after HCV SVR was 6.44 (95% CI 2.49–16.69) per 100PY in a meta-analysis studied by Aspinall et al. [21]. The impact of ongoing injected drug use post-SVR or the end of HCV therapy was presented in further studies. Midgard et al. showed that the reinfection incidence rose to 5.8 per 100PY (95% CI 3.0–10.2) among participants who reported injected drug use post-treatment, compared with those who abstained from injecting for more than six months (2.0 per 100PY (95% CI 1.0–3.5)) [17]. Weir et al. also demonstrated that the incidence of reinfection was 5.7 per 100PY (95% CI 1.8–13.3) among PWID who had been hospitalized for an opiate- or injection-related cause post-SVR, and the overall reinfection incidence was 1.7 per 100PY (95% CI 0.7–3.5) among lifetime-long PWID following SVR in Scotland [22]. In our study, the adjusted hazard ratio of heroin dependency in PWID was 7.41 (95% CI 3.37–14.3). Compared with those in the previous studies, our observational cohort study produced a similar result, but our data contributed toward countering the lack of information on Asian subjects.

Research beyond this has demonstrated how methadone replacement therapy can reduce the incidence of HCV reinfection in PWID. Hajarizadeh et al. found that the HCV reinfection risk following treatment was higher in people with recent drug use who were not receiving OST (6.6/100PY) than those receiving OST with (5.9/100PY) or without recent drug use (1.4/100PY) [13]. A meta-analysis study showed the pooled reinfection incidence was 2.4 per 100PY (95% CI 0.9–6.1) in PWID, but the incidence increased up to 6.4 per 100PY (95% CI 2.5–16.7) in heroin-dependent people who did not receive OST [21]. Our study showed a similar trend (AHR, 0.62; 95% CI 0.34–1.14), but the result was not significant due to our potential underestimation of the number of heroin-dependent PWID receiving methadone replacement therapy.

Among PWID who completed HCV therapy with DAAs, our study showed a similar incidence of reinfection (6.01/100PY) to that of the C-EDGE COSTAR trial (DAA-based treatment), which identified 11 cases of reinfection after the end of treatment, with an incidence of 4.6 per 100PY (95% CI 1.7–10.0) [22]. The first DAA treatment was launched in 2016 in Taiwan, and the scaling-up of easy and less-toxic DAA treatments led to substantial reductions in the HCV prevalence. However, this also brought about a high incidence of reinfection due to the convenient regimen and short treatment duration, compared with the treatment regimen of peginterferon in PWID who either have HIV (an incidence of 3.2 per 100PY) or do not (an incidence of 2.23 per 100PY) [23].

Liver cirrhosis was also a risk factor of reinfection in our study, with an incidence of 18.1/100PY noted (AHR, 3.94; 95% CI 1.09–14.22), though we could not distinguish reinfection from relapse due to a lack of phylogenetic analysis of HCV strains. Nonetheless, Pawel et al. demonstrated significantly lower response rates in patients with liver cirrhosis in their research (odds ratio, 4.43; 95% CI, 1.59–12.38) [24].

A high incidence of HCV reinfection in MSM living with HIV has also been reported in several European countries since 2011, with incidence rates ranging from 4.8 to 21.8 per 100PY [9,11,16,25], and our study showed an incidence of 5.8 per 100PY. In combination, having multiple sexual partners and abusing amphetamines might increase the risk of reinfection (AHR: 20.17; 95% CI, 2.36–172.52). We found that all subjects who had multiple sexual partners abused amphetamines when they were diagnosed with HCV reinfection. The potential mode of transmission is anal intercourse with risk of blood–blood contact, and abuse of amphetamine was the prime factor to put men at high risk in this way of infection with HCV or other sexually transmitted infections [26].

Taiwan CDC published an HIV pre-exposure prophylaxis (PrEP) guideline and initiated a government-led PrEP pilot targeting 1000 at-risk individuals from November 2016 to August 2017 [27]. For this reason, the HCV treatment period was analyzed before and after 2017; however, we found that the incidence from 2010 to 2016 (6.7/100PY) was not significantly different from that from 2017 to 2019 (5.0/100PY; AHR, 0.83, 95% CI 0.17–4.02). We speculated that a high treatment uptake and early diagnosis likely suppressed the reinfection rates in our study, regardless of the activity of the PrEP pilot.

There are some limitations to our study that we must note here before concluding. First, the retrospective and observational nature of the study may have meant we missed out on detailed patient information, and this could have limited the analysis of potential risks. Second, some patient accounts may have been affected by recall bias or they may have concealed detailed behavioral information, such as heroin abuse, needle sharing, injection of amphetamine, or a high number of sexual partners, which may have led us to underestimate the effect of risky behavior. Third, HCV RNA PCR testing was not carried out at standardized intervals; instead, this depended on the physician’s decision or the patient’s medical condition. The estimated date of reinfection was the day of the first positive test result, but the exact date might have been at any time between the last negative and first positive test results. Therefore, we may have underestimated the incidence of reinfection. Fourth, we could not distinguish reinfection from late relapse as we did not perform a phylogenetic analysis of the HCV strains. Finally, this single-center study may not be generalizable to other hospitals or countries given the patient differences of various demographics and clinical characteristics between one location and another.

## 5. Conclusions

In conclusion, our study revealed an incidence of HCV reinfection similar to that found in Europe, which was significantly associated with risky behaviors, including heroin dependency and amphetamine abuse. Hence, we propose a need for education and behavioral interventions to prevent reinfection in this population.

## Figures and Tables

**Figure 1 viruses-14-00439-f001:**
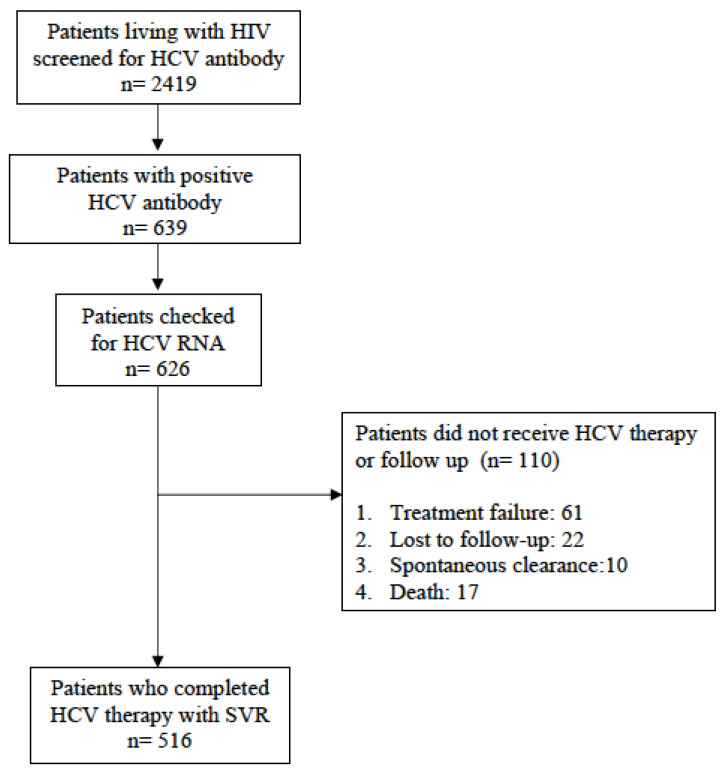
Flowchart of enrollment of study population since 2009.

**Figure 2 viruses-14-00439-f002:**
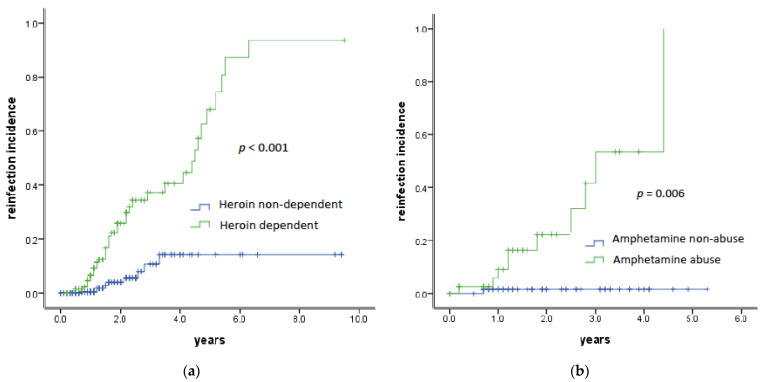
Incidence of HCV reinfection in subgroups of PWID (**a**) and MSM (**b**) with heroin dependency or amphetamine abuse by Kaplan–Meier curves.

**Table 1 viruses-14-00439-t001:** Characteristics of HIV-positive patients with or without hepatitis C virus (HCV) reinfection at baseline.

	Total (n = 516)	Reinfected (n = 59)	Not Reinfected (n = 457)	*p*
age, year, mean (SD)	42 (9)	40.1 (7.7)	42.4 (8.6)	0.067
male, n (%)	407 (78.9)	58 (98.3)	349 (76.4)	<0.001
PWID, n (%)	407 (78.9)	48 (81.4)	359 (78.6)	0.718
MSM, n (%)	105 (20.3)	11 (18.6)	94 (20.6)	0.571
CD4 count in HCV treatment, cells/μL, median (IQR)	506 (369–692)	475 (360–701)	514 (369–690)	0.3
HIV viral loads <50 copies/mL in HCV treatment, n (%)	463 (89.7)	49 (83.1)	414 (90.6)	0.073
HCV RNA before treatment, log_10_, median (IQR)	6.32 (5.5–6.8)	6.37 (5.1–6.9)	6.32 (5.6–6.8)	0.908
HCV genotype				
1a, n (%)	101 (19.6)	14 (23.7)	87 (19.0)	0.393
1b, n (%)	117 (22.7)	17 (28.8)	100 (21.9)	0.231
2, n (%)	82 (15.9)	9 (15.3)	73 (16.0)	0.887
3, n (%)	44 (8.5)	8 (13.6)	36 (7.9)	0.141
6, n (%)	176 (34.1)	11 (18.6)	165 (36.1)	0.008
HCV treatment				<0.001
DAA (direct-acting antiviral), n (%)	387 (75)	27 (45.8)	360 (78.8)	
Peginterferon ± ribavirin, n (%)	129 (25)	32 (54.2)	97 (21.2)	
HCV treatment year				<0.001
2010–2016, n (%)	106 (21)	29 (49.2)	77 (16.8)	
2017–2019, n (%)	410 (79)	30 (50.8)	380 (83.2)	
Liver cirrhosis, n (%)	17 (3.3)	3 (5.1)	14 (3.1)	0.414

Footnote: PWID, people who inject drugs; MSM, men who have sex with men.

**Table 2 viruses-14-00439-t002:** Counseling interview on risky behavior and immunological status at follow-up.

	Total (n = 516)	Reinfected (n = 59)	Not Reinfected (n = 457)	*p*
Follow-up time after treatment, weeks, median (IQR)	63.6 (48.3–114.5)	85.3 (57.0–150)	62.4 (47.6–106)	0.009
Risky behavior and disease				
Heroin dependency, n (%)	142 (27.5)	39 (66.1)	103 (22.5)	<0.001
Methadone replacement therapy, n (%)	167 (32.4)	23 (39.0)	144 (31.5)	0.248
Amphetamine abuse, n (%)	71 (13.8)	11 (18.6)	60 (13.1)	0.247
Multiple sexual partners, n (%)	38 (7.4)	10 (16.9)	28 (6.1)	0.003
Recent syphilis, n (%)	35 (6.8)	5 (8.5)	30 (6.6)	0.583
No risky behavior, n (%)	248 (48.1)	6 (10.2)	242 (53.0)	<0.001
CD4 count at follow-up, cells/μL, median (IQR)	550 (414–764)	480 (377–652)	573 (423–779)	0.022
HIV viral loads <50 copies/mL at follow-up, n (%)	466 (90.3)	47 (79.7)	419 (91.7)	0.003

**Table 3 viruses-14-00439-t003:** Risk factors and reported behavior at the end of the follow-up or at hepatitis C virus (HCV) reinfection in Cox proportional hazards regression model.

	Crude Hazard Ratio	95% CI	*p*	Adjusted Hazard Ratio	95% CI	*p*
Male	11.33	1.57–81.95	0.016	6.7	0.91–49.39	0.062
HCV genotype 6	0.69	0.35–1.33	0.262	0.74	0.37–1.47	0.387
DAA (direct-acting antiviral)	0.61	0.31–1.19	0.145	1.7	0.49–5.91	0.401
Treatment year 2017 to 2019	0.57	0.29–1.14	0.111	1.31	0.36–4.8	0.68
Liver cirrhosis	5.99	1.78–20.15	0.004	3.61	1.02–12.71	0.046
Heroin dependency	4.32	2.51–7.44	<0.001	9.35	4.28–20.44	<0.001
Methadone replacement therapy	0.72	0.43–1.22	0.223	0.64	0.34–1.18	0.151
Amphetamine abuse	1.78	0.92–3.46	0.088	0.16	0.02–1.61	0.07
Multiple sexual partners	3.5	1.75–7.00	<0.001	56.37	6.12–519.53	<0.001
Recent syphilis	1.1	0.44–2.77	0.836	1.11	0.38–3.31	0.846
CD4 count >500 cells/μL at follow-up	1.92	1.14–3.22	0.014	0.75	0.41–1.35	0.33
HIV viral load <50 copies/mL at follow-up	0.59	0.31–1.11	0.099	0.58	0.3–1.12	0.104

**Table 4 viruses-14-00439-t004:** Risk factors and reported behavior at the end of the follow-up or at hepatitis C virus (HCV) reinfection in the subgroup of PWID.

	Crude Hazard Ratio	95% CI	*p*	Adjusted Hazard Ratio	95% CI	*p*
Male	12.14	1.67–88.09	0.014	8.02	1.08–59.49	0.042
HCV genotype 6	0.68	0.35–1.35	0.272	0.78	0.39–1.55	0.473
DAA (direct-acting antiviral)	2.07	0.93–4.61	0.073	2.23	1.04–4.79	0.04
Liver cirrhosis	6.62	1.92–22.92	0.003	3.94	1.09–14.22	0.037
Heroin dependency	6.55	3.17–13.55	<0.001	7.41	3.37–16.3	<0.001
Methadone replacement therapy	1.37	0.78–2.42	0.276	0.62	0.34–1.14	0.126
Amphetamine abuse	0.30	0.04–2.15	0.229	0.15	0.02–1.13	0.066
CD4 count >500 cells/μL at follow-up	0.58	0.33–1.04	0.068	0.93	0.5–1.74	0.82
HIV viral load <50 copies/mL at follow-up	0.46	0.24–0.88	0.019	0.47	0.24–0.93	0.031

**Table 5 viruses-14-00439-t005:** Risk factors and reported behavior at the end of the follow-up or at hepatitis C virus (HCV) reinfection in the subgroup of MSM.

	Crude Hazard Ratio	95% CI	*p*	Adjusted Hazard Ratio	95% CI	*p*
Treatment year 2017 to 2019	1.07	0.26–4.47	0.925	0.83	0.17–4.02	0.813
Amphetamine abuse	21.57	2.74–169.79	0.004	20.17	2.36–172.52	0.006
Recent syphilis	1.31	0.39–4.41	0.667	0.8	0.2–3.15	0.749
CD4 count >500 cells/μL at follow-up	0.32	0.09–1.09	0.067	0.66	0.14–3.1	0.596

## Data Availability

All data analyzed or generated during this study are included in the manuscript.

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
