# Peer review of "Incidence and Risk Factors of Reinfection with HCV after Treatment in People Living with HIV"

_viruses, 2022, doi:10.3390/v14020439_

Round 1

Reviewer 1 Report

Overall interesting study. Even though similar studies have been reported, there seems to be little information in this particular population.

General remarks:

  • “Person-before disease” language should be used consistenty (in particular person/people living with HIV instead of HIV-infected people).
  • The manuscript might in general profit from language and style polishing as in parts it is a bit difficult to read. This is particularly true given the many groups and subgroups that were analyzed making it sometimes very hard for a reader to follow.

  • The observation period is rather long and many things might have changed that could be of potential influence on the recurrence rate of HCV one of which is the more widespread use of PREP that might have led to more unprotected sex. Could the authors think about a method to implement the time as an (autoregressive) variable into the model to adjust for that or try to present reinfection incidences stratified by time (years) in a meaningful way. Otherwise, this must be discussed at least.
  • There is no mentioning of people who were found to be antibody positive but did not have a positive HCV-PCR. Did this not occur or how were these patients managed for data analysis?
  • Is there a reason for excluding those with spontaneous clearance of HIV from the analysis? As re-infection could also be assessed in these people there seems no proper reason for me to do so (except for the title of the manuscript ?)
  • IQRs are given as a single “number” which is technically correct when defining the IQR as the span between the lower and the upper quartile. However, it is more common and more informative to give the upper and lower limits as this gives a better understanding of the spread of the data around the median as for data for which the median is used, an asymmetric (or at least not normal) distribution can be assumed.
  • Line 123: Cis were computed using a binomial distribution > I think this refers to the 95%CI of the HRs, which has not been mentioned by then. This should be clarified.
  • There is no mentioning of the model-building method used for the proportional hazards model (was a model actually fit (and if so, by which criteria) or was the time “only” adjusted for all the variables the authors mentioned? In the later case there should be a justification for the selection of these variables) and no mentioning if model assumptions were checked and fulfilled.
  • Can the authors explain, why recent syphilis was considered to be an assumedly independent risk factor for HIV-re-infection? To me it is questionable if a recent syphilis infection is truly an independent variable or maybe more an intermediate variable that might have considerable collinearity with multiple sex partners and should therefore be eliminated from the model (this, again, refers to the problem of missing information on the model building strategy in the previous comment).
  • Table 4: it might not be intuitive why DAA treatment seems to have higher HRs of re-infection in all risk categories. Could this be a time effect or a selection bias? Same for the results when stratified by CD4 cell count. Could the authors please discuss these findings.
  • Table 4: I think the last row of “results” being all equal to one could be eliminated as this always holds for the reference category.
  • Lines 266-269: What do the authors mean by reinfections are most likely suppressed? Could you please elaborate on this statement or delete it. Same is true for the following sentence: As the authors present before that according to their result there is no significant difference in re-infection for people treated with INF and DAA, it is hard to understand why in the next sentence they highlight the need for DAAs (which I support but which seems odd in this context).
  • References: Formatting is inconsistent. Some authors are presented as “first name, last name” while others are presented (in the preferable way) “last name, first name”. Also reference No. 5 is just the second part of reference No. 4 which also makes the numbering in the manuscript wrong, which included (of course), that there is no reference 26 in the manuscript but in the list of references.

Author Response

Dear reviewer,

We would like to thank you and the reviewers for giving us the opportunity to improve our manuscript entitled “Incidence and risk factors of re-infection of HCV after treatment in people living with HIV”. Our point-to-point response to the reviewers' comments and queries are listed below.

Reviewer 1:

  1. “Person-before disease” language should be used consistenty (in particular person/people living with HIV instead of HIV-infected people).

Answer: Thank for your suggestion, and we corrected this terminology as people living with HIV (PLWH).

  1. The manuscript might in general profit from language and style polishing as in parts it is a bit difficult to read. This is particularly true given the many groups and subgroups that were analyzed making it sometimes very hard for a reader to follow.

Answer: Thank for your suggestion, we simplified our study population in only two subgroups of PWID and MSM in table 3a, 3b and 3c, and deleted table 4. We hope it will be much easier for readers

  1. The observation period is rather long and many things might have changed that could be of potential influence on the recurrence rate of HCV one of which is the more widespread use of PREP that might have led to more unprotected sex. Could the authors think about a method to implement the time as an (autoregressive) variable into the model to adjust for that or try to present reinfection incidences stratified by time (years) in a meaningful way. Otherwise, this must be discussed at least.

Answer: We added one more variable of treatment period “from 2010 to 2016” and “from 2017 to 2019” due to HIV PrEP program initiated since 2016 October. In table 1 and table 3c, we analyzed the outcome, but there was no significant difference. The possible reason was a higher treatment uptake and earlier diagnosis likely suppressed reinfections rate in our study. (line 326 - 336)

  1. There is no mentioning of people who were found to be antibody positive but did not have a positive HCV-PCR. Did this not occur or how were these patients managed for data analysis?

Answer: Thank for your remind, and we did find out some people with positive HCV antibody, but HCV-PCR was undetectable, and the proportion was less than 2% (10 patients) in our population. We followed up their AST/ALT at outpatient clinics, and if AST/ALT was abnormal, we would re-check HCV-RNA, but if ASL/ALT was, we could not re-check HCV-PCR according to the Taiwan National Health Insurance’s regulation.  

  1. Is there a reason for excluding those with spontaneous clearance of HIV from the analysis? As re-infection could also be assessed in these people there seems no proper reason for me to do so (except for the title of the manuscript)

Answer: We didn’t include those with spontaneous clearance of HCV infection in our study, but if their HCV-PCR became detectable during the follow-up, and they completed the treatment, then we would include this population in our observational study.

  1. IQRs are given as a single “number” which is technically correct when defining the IQR as the span between the lower and the upper quartile. However, it is more common and more informative to give the upper and lower limits as this gives a better understanding of the spread of the data around the median as for data for which the median is used, an asymmetric (or at least not normal) distribution can be assumed.

Answer: Thank for your remind, and we corrected IQR as your suggestion.

  1. Line 123: Cis were computed using a binomial distribution > I think this refers to the 95%CI of the HRs, which has not been mentioned by then. This should be clarified.

Answer: The 95% CI of hazard ratio was computed using a binomial distribution.

  1. There is no mentioning of the model-building method used for the proportional hazards model (was a model actually fit (and if so, by which criteria) or was the time “only” adjusted for all the variables the authors mentioned? In the later case there should be a justification for the selection of these variables) and no mentioning if model assumptions were checked and fulfilled.

Answer: The model building method used for our study was the time “only” adjusted for all the variables. Some important variables (a p < 0.1 in χ2, Fisher’s exact tests or univariate analysis) were selected for subsequent multivariate analysis.

  1. Can the authors explain, why recent syphilis was considered to be an assumedly independent risk factor for HCV-re-infection? To me it is questionable if a recent syphilis infection is truly an independent variable or maybe more an intermediate variable that might have considerable collinearity with multiple sex partners and should therefore be eliminated from the model (this, again, refers to the problem of missing information on the model building strategy in the previous comment).

Answer: Thank for your remind, we re-checked other variables, and found recent syphilis infection was an intermediate variable, and the prime risk factor was amphetamine abuse in subgroup of MSM, and we showed the result in table 3c.

  1. Table 4: it might not be intuitive why DAA treatment seems to have higher HRs of re-infection in all risk categories. Could this be a time effect or a selection bias? Same for the results when stratified by CD4 cell count. Could the authors please discuss these findings.

Answer: Thank for your suggestion, we simplified our study population in only two subgroups of PWID and MSM in table 3a, 3b and 3c, and deleted table 4. We hope it will be much easier for readers

  1. Table 4: I think the last row of “results” being all equal to one could be eliminated as this always holds for the reference category.

Answer: Thank for your suggestion, we simplified our study population in only two subgroups of PWID and MSM in table 3a, 3b and 3c, and deleted table 4.

  1. Lines 266-269: What do the authors mean by reinfections are most likely suppressed? Could you please elaborate on this statement or delete it. Same is true for the following sentence: As the authors present before that according to their result there is no significant difference in re-infection for people treated with INF and DAA, it is hard to understand why in the next sentence they highlight the need for DAAs (which I support but which seems odd in this context).

Answer: We added one variable of treatment year, and deleted one variable of multiple sex partners, because this could be also an intermediate variable. Finally, we deleted the variable of DAA in subgroup of MSM, because there was no significant difference.

  1. References: Formatting is inconsistent. Some authors are presented as “first name, last name” while others are presented (in the preferable way) “last name, first name”. Also reference No. 5 is just the second part of reference No. 4 which also makes the numbering in the manuscript wrong, which included (of course), that there is no reference 26 in the manuscript but in the list of references.

Answer: Thank for your correction, and we adjusted the format of reference.

Reviewer 2 Report

Please see attached comments.

Author Response

Dear reviewer,

We would like to thank you and the reviewers for giving us the opportunity to improve our manuscript entitled “Incidence and risk factors of re-infection of HCV after treatment in people living with HIV”. Our point-to-point response to the reviewers' comments and queries are listed below.

Reviewer 2:

Thank you for the opportunity to review this paper on HCV among people with HIV. This manuscript provides detailed information on HCV reinfection among a population living with HIV. While the paper adds valuable information to the literature, significant editing, additional context, and more thorough description of the issue need to be included. Overall Be careful with your use of terminology.

  1. The term HIV positive patient is not appropriate. It is better to use person first language – People/person with HIV is a preferred term to use. Please make sure to use that throughout the manuscript.

Answer: Thank for your suggestion, and we corrected this terminology as people living with HIV (PLWH).

  1. Abstract The second sentence is not a sentence – it is missing a verb. “increasing incidence…has been described”? This is a retrospective cohort study conduct from 2009 to 2020 – but from where? Both location and type of facility (hospital system, community clinic, academic medical center, etc) are needed. Be sure to define any acronyms prior to using them (PWID, MSM are not defined).

Answer: This sentence was adjusted as below. Increasing incidence of acute hepatitis C infection in people living with HIV (PLWH) was described in recent years. This retrospective cohort study was conducted in PLWH who completed HCV therapy between 2009 June and 2020 June in one single HIV care designed hospital.

  1. The third from last (line 25) and second from last (line 27) are redundant.

Answer: Line 27 was deleted, thank you.

  1. Introduction The introduction is very choppy and difficult to follow. The second paragraph needs an introductory sentence that provides an overview of the content of the paragraph, not data. A consultation with a writing editor may help the overall flow and English grammar of this paper.

Answer: Thank for your suggestion, and we consulted with a native English writer, and apology for our poor writing.    

  1. The last sentence/purpose statement (lines 59-62) may need to be broken down into 2 sentences or delineated as such. It is difficult to follow as written.

Answer: Thank for your correction, and the sentence was revised as below. Hence, we assessed the incidence of HCV reinfection after sustained virological response and examined behavioral and immunological risk factors in the observational cohort study.

  1. Materials and Methods The first paragraph only has 2 sentences. The second sentence is a run-on sentence that needs to be broken down. The variables that you captured are important. As written, it is almost impossible to understand what they are. Line 78, a systemic procedure for HCV infection… how often was this done? Every 6 months? Every year? Why were transgender individuals not included in this analysis? Risk behavior information from the case manager is not clear. Was this extracted from the EHR? If these counseling appointments were every 3 months, what time point was used to extract this data? What if risk behaviors changed?

Answer: Thank for your correction, and the sentence was revised. A systemic procedure for HCV infection was done annually, and there was only 3 transgender individuals in our hospital, and they had negative HCV Ab. In Taiwan, face-to-face interviews in assessing illicit drug abuse (such as heroin, amphetamine and methadone) and with sexual risk behavior were performed by HIV case managers, when patients returned for refilling highly active anti-retroviral therapy every three months at outpatient clinics. If acute HCV infection was diagnosed during the follow-up, clinical covariates were taken as the closest measurement to the date of a follow-up, including CD4 lymphocyte count and HIV RNA viral loads, HCV RNA viral load, HCV genotype, regimens of HCV treatment and STDs, and risky behavior was assumed over the past 3 months.   

  1. In the analysis section, line 127, the authors state that some important variables were selected for multivariate analysis. How were those variables chosen? Please provide more detail.

Answer: Some important variables (a p < 0.1 in χ2, Fisher’s exact tests or univariate analysis) were selected for subsequent multivariate analysis.

  1. Results It is unclear why tables 1 and 2 are separate. They could be combined into one demographics table

Answer: Table 1 shows the data at screening, and table 2 showed the data or information at end of follow-up or time of HCV reinfection. 

  1. Table 4 includes selected combinations. Where are the results of other combinations? Where are the results of the full model (i.e., all possible combinations)? If your outcome is HCV reinfection (y/n), this analysis should be described as multivariate, not multivariable.

Answer: Thank for your suggestion, we simplified our study population in only two subgroups of PWID and MSM in table 3a, 3b and 3c, and deleted table 4.

  1. Discussion Please apply your results to the context of other studies more thoroughly instead of just listing the results of the other studies. Help the reader understand how your results fit in with the existing literature and what your results add (paragraph 2) It would be useful to provide context either in the introduction or the discussion as to why HCV reinfection is an issue. What are the health consequences of HCV/HIV co-infection?

Answer: Thank for your important suggestion, and we adjusted the content and hope reader might understand those information.

  1. Line 257, what do you mean traumatic sexual behavior? If the authors mean anal intercourse, please state that.

Answer: thank you, we corrected this term.

  1. Line 261 – 263, I don’t understand how reinfection or well controlled HIV correlate with awareness of protective sex. Also, those do not correlate at all with willingness to have protective sex. Be careful with broad, sweeping statements.

Answer: Thank for your suggestion, we deleted this sentence.

  1. Line 268 – MSM require close behavioral interventions…what does that mean? Please provide more context to this statement. What types of behavioral interventions do the authors think would be beneficial to prevent HCV reinfection among MSM with HIV?

Answer: We corrected this sentence, and suggested that if people may have abstinence from amphetamine abuse, the risk of HCV reinfection may decrease.

  1. Same with more convenient tests – do the authors mean cheaper, less cumbersome, self-tests, etc?

Answer: Also, we revised “Discussion”, and hope you understand our content, thank you.
